# Subunit gating resulting from individual protonation events in Kir2 channels

Grigory Maksaev [1,3], Michael Bründl-Jirout [2,3], Anna Stary-Weinzinger [2], Eva-Maria Zangerl-Plessl [2] ✉, Sun-Joo Lee [1] & Colin G. Nichols [1] ✉

Inwardly rectifying potassium (Kir) channels open at the 'helix bundle crossing' (HBC), formed by the M2 helices at the cytoplasmic end of the transmembrane pore. Introduced negative charges at the HBC (G178D) in Kir2.2 channels forces opening, allowing pore wetting and free movement of permeant ions between the cytoplasm and the inner cavity. Single-channel recordings reveal striking, pH-dependent, subconductance behaviors in G178D (or G178E and equivalent Kir2.1[G177E]) mutant channels, with well-resolved non-cooperative subconductance levels. Decreasing cytoplasmic pH shifts the probability towards lower conductance levels. Molecular dynamics simulations show how protonation of Kir2.2[G178D], or the D173 pore-lining residues, changes solvation, $K^+$ ion occupancy, and $K^+$ conductance. Ion channel gating and conductance are classically understood as separate processes. The present data reveal how individual protonation events change the electrostatic microenvironment of the pore, resulting in step-wise alterations of ion pooling, and hence conductance, that appear as 'gated' substates.

Potassium channels are present in all types of cells and play critical roles in control of multiple physiological processes. In contrast to voltage-gated channels (such as $K_V$), inward rectifier potassium channels (Kir)[1] lack the four-helix voltage sensing domain, although classical strong inward rectifying (Kir2 sub-family) channels do demonstrate a voltage dependence arising from block by intracellular $Mg^{2+}$ and positively charged polyamines[2,3]. Kir channels consist of only two transmembrane helices plus an extensive C-terminal cytoplasmic tail of the channel pore. All Kir channels require the regulatory ligand phosphatidylinositol 4,5-bisphoshate ($PIP_2$)[4,5], as well as a number of subfamily-specific ligands, such as $G_{\beta\gamma}$ proteins, ADP/ATP, sulphonylurea receptor subunits (SURs), and pH[3] for normal activity. These agents generally act through interaction with the cytoplasmic domain (CTD), as do bulk anionic lipids, such as phosphatidylglycerol, and phosphatidylserine[6,7], which significantly increase $PIP_2$ sensitivity of Kir2.x channels.

Traditional crystallography and cryo-EM techniques have now provided multiple structures for members of the classical strong

inward rectifier Kir2 subfamily[5,8–10]. These structures show many relevant features but still do not capture the full ensemble of functional conformations. In particular, most structures to-date represent closed conformations in which the narrowest part of the conductive pathway, located at the cytoplasmic membrane interface and formed by the M2 helix bundle crossing (HBC), provides a major hydrophobic bottleneck that blocks ion conduction[5,9]. This implicates expansion of the HBC as a critical step in Kir channel opening, as seen in a recent Kir6.2 structure[11]. We reported the crystal structure of a chicken Kir2.2[G178D] mutant channel[12], in which the introduced G178D mutations at the HBC functionally stabilize the open conformation, a strategy used previously to obtain an open crystal structure of a bacterial homolog KirBac3.1[13]. In the G178D structure, the HBC gate is slightly wider than in previous structures, and molecular dynamics (MD) simulations demonstrate rapid wetting of the G178D pore at the HBC region, followed by further expansion and $K^+$ conductance through the channel.

[1]Department of Cell Biology and Physiology, and the Center for Investigation of Membrane Excitability Diseases, Washington University School of Medicine, St. Louis, Missouri, USA. [2]Department of Pharmaceutical Sciences, Division of Pharmacology and Toxicology, University of Vienna, Vienna, Austria. [3]These authors contributed equally: Grigory Maksaev, Michael Bründl-Jirout. ✉e-mail: eva-maria.zangerl@univie.ac.at; cnichols@wustl.edu

In the present study, we carry out detailed single-channel analyses of cKir2.2[G178D] and hKir2.1[G177E] channels that reveal striking substate gating behavior. These channels show multiple, stochastically distributed, subconductances, that are quite different from the occasional subconductances that have been reported for wild type Kir2.1 or Kir2.2[14,15]. While subconductance gating has been sporadically observed in many related and unrelated channels[14–18], poor resolution and inconsistency of experimental findings has rarely illuminated the underlying basis. In the present study, the behavior is very consistent. Occupancy of the sub-conducting states is a function of the cytoplasmic pH, favoring the more conductive sub-states at higher pH. MD simulations and additional mutagenesis experiments indicate that this remarkable subunit-dependent gating behavior is explained by each subconductance state being a direct consequence of protonation of individual negatively charged side chains within the channel pore.

## Results

### Sub-conducting states in Kir2.2 G178D and G178E channels

We previously reported a 'forced open' Kir2.2 channel with a single residue mutation G178D, located right at the HBC[12] (Fig. 1A). This mutant demonstrated activity in the absence of the normally obligatory ligand PI(4,5)P$_2$, as well as an increased unitary conductance (~60 pS vs. ~46 pS for WT) in excised patches[12]. Closer analysis reveals subconductances within the single channel current (Fig. 1B) that are completely absent in WT Kir2.2 currents (Fig. 2B and Supplementary Fig. 1). Under symmetric pH 7.4 buffer conditions we identified four distinct conducting states (labeled $O_1$-$O_4$) with relative amplitudes of 0.45, 0.74, 0.92 and 1.00 (Fig. 1B). Although the fully open state still

dominated at physiological pH ($P^{O4} = 0.59$), the other sub-states contributed significantly: $P^{O3} = 0.26$, $P^{O2} = 0.07$ and $P^{O1} = 0.02$ (Supplementary Table 1). The vast majority of transitions occurred between adjacent conducting levels, although transitions to and from the zero-conductance level (C) were more frequent than transitions between non-adjacent conducting levels (Fig. 1C, Supplementary Table 1). The similar mutation G178E also demonstrated increased unitary conductance and four open states (Fig. 1D) that were almost identical to the G178D conducting states, indicating that the appearance of this sub-state gating was directly caused by replacement of glycine at position 178 with an acidic residue.

### Selectivity filter gating is unaffected by the G178D mutation

We further analyzed the brief intra-burst closures (burst closure cutoff <100 msecs) that occur in both WT and G178D channels, at pH 7.4. In the case of the mutant, only closures after which the channel returned to the sub-state that it closed from (i.e., $O_i \rightarrow C \rightarrow O_i$) were used for analysis. The frequency of occurrence was slightly lower for G178D ($3.7\,s^{-1}$) than for WT ($5.9\,s^{-1}$), potentially due to exclusion of $O_i \rightarrow C$ events that apparently returned to states other than $O_i$ in G178D, but C state dwell times ($6.4 \pm 0.5$ ms and $7.1 \pm 0.5$ ms, mean $\pm$ S.E.) were not significantly different between WT and G178D ($p = 0.343$, unpaired $T$-test). Furthermore, the fraction of transitions between the fully closed state (C) and any of the conducting sub-states ($O_1$ to $O_4$) in G178D (Fig. 1C) followed the probability distribution of these sub-states (Supplementary Table 1). Therefore, we suggest the transitions to the C state are distinct closures (potentially at the selectivity filter) that occur independently of the G178D-driven, stepwise sub-state

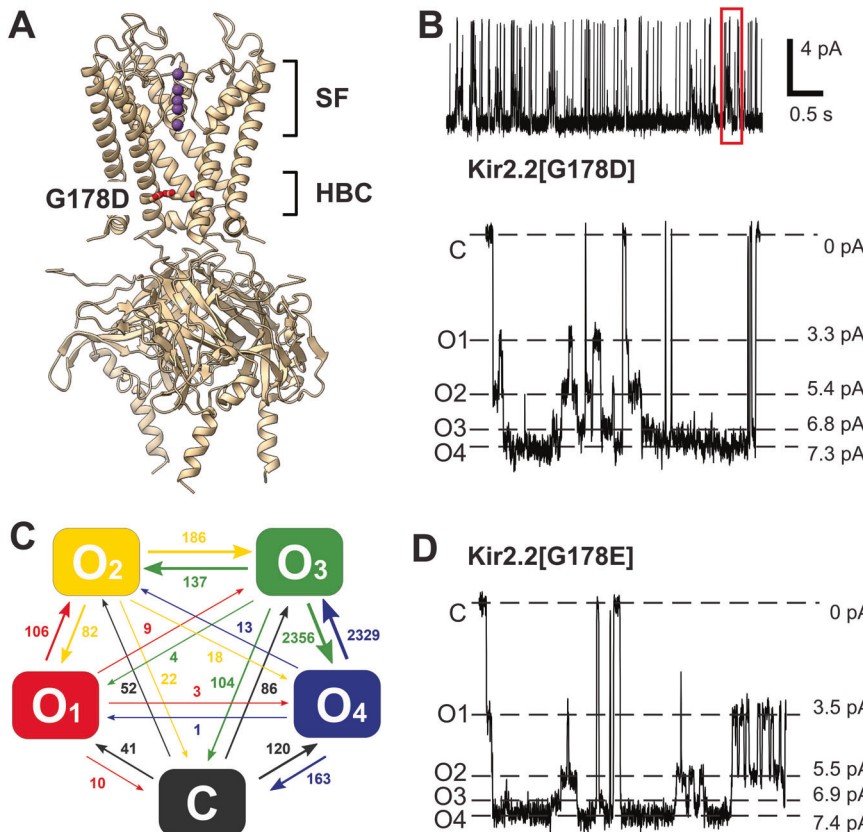

**Fig. 1 | Sub-state conductance in cKir2.2[G178D] at pH 7.4. A** Ribbon diagram indicating location of Kir2.2[G178D] near the helix bundle crossing (HBC) and the selectivity filter (PDB ID: 6M86). Only 3 subunits out of four are shown for clarity. **B** Representative 5 s long trace of Kir2.2[G178D] channel activity (top panel) recorded in symmetric 150 mM KCl at pH 7.4. (*Bottom panel*) Closed (C), completely open (O4) and 3 unevenly spaced subconductance states (O1-O3) resolved

at −120 mV membrane. **C** Summary of total transitions between conducting substates in cKir2.2[G178D] mutant at pH 7.4, from a 20 s long recording. The vast majority of transitions occurs between adjacent levels, i.e., the outer transitions. **D** Representative trace of Kir2.2[G178E] channel activity: a closed (C), a completely open (C4) and 3 unevenly spaced subconductance states (C1-C3) resolved at −120 mV membrane.

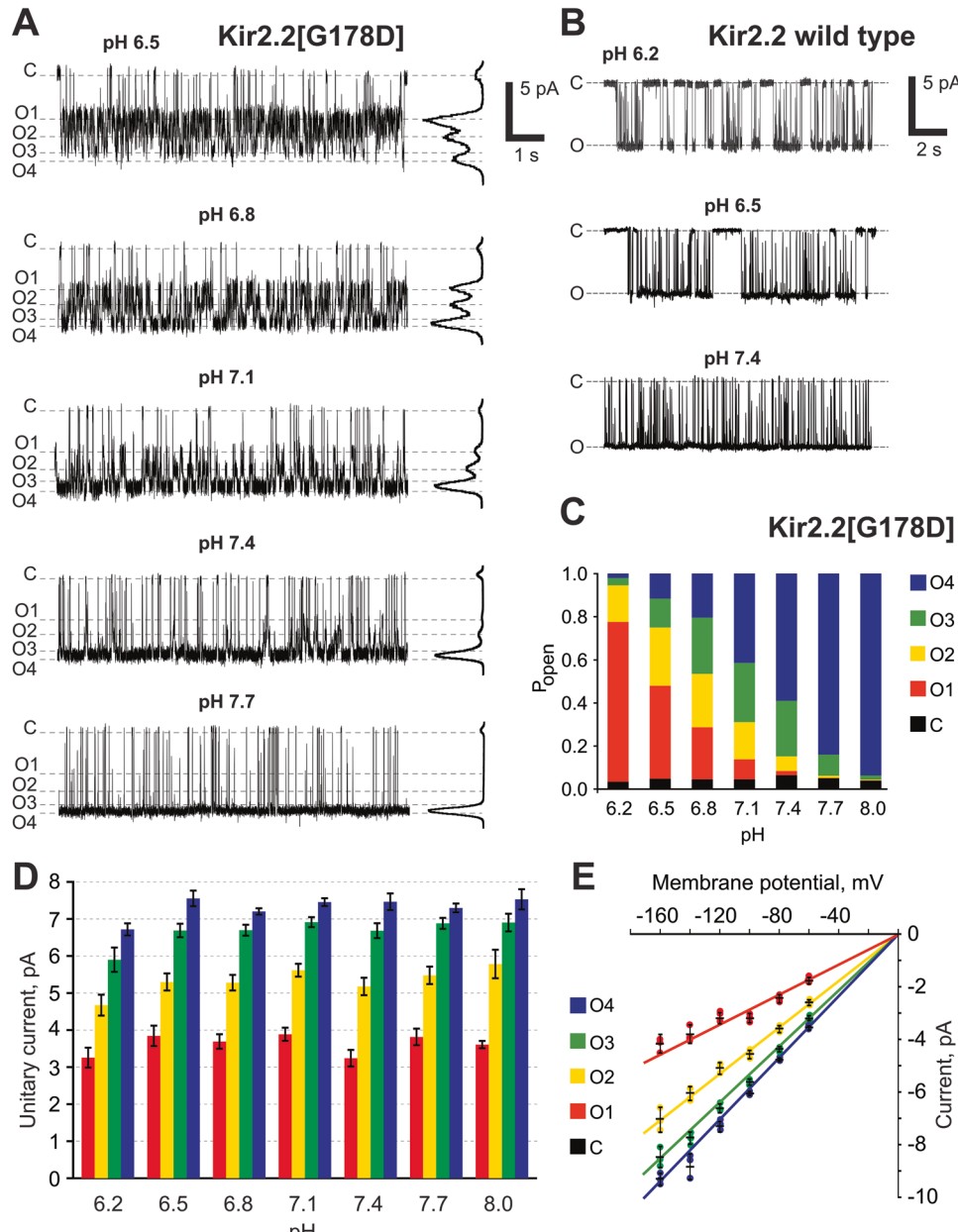

**Fig. 2 | pH-dependence of cKir2.2[G178D] sub-states. A** Representative traces of Kir2.2[G178D] channel activity at different pH. All-point histograms are presented on the right of each trace. **B** Representative traces of WT Kir2.2 channel activity at different pH. **C** Distribution of the open probabilities of conductive states of cKir2.2[G178D] at different pH levels. Data are taken from **A. D** Sub-states amplitudes of cKir2.2[G178D] at −120 mV membrane potential at different pH. **E** Sub-state amplitudes of cKir2.2[G178D] at pH 7.4 at different membrane potentials. Sub-states could not be reliably resolved at potentials more positive than −60 mV due to a low signal to noise ratio. Data in **D** and **E** shown as mean ± s.e.m. ($n$ = 50–1000 transitions within each patch, $N$ = 3–5 patches, for each conductance level).

transitions. We further suggest that the rare transitions between non-adjacent conducting states in Fig. 1C, are likely to represent combinations of sequential transitions between adjacent conducting levels that are not temporally resolved at the experimental data acquisition and hardware filter frequencies (3 kHz / 1 kHz, see Methods for details).

## Distribution of sub-conducting states is pH-dependent and voltage-independent

While the pKa of solvent-exposed aspartate side chains is very acidic (~pKa 3), it can be strongly modulated by the protein and membrane microenvironment[19–21], and has been very extensively documented for protonatable side chains in the pore of nicotinic acetylcholine receptors and other Cys-loop channels[22,23]. Thus, it is conceivable that the

effective pKa may shift within the channel pore, and that the sub-states are a consequence of G178D side chain protonation. We therefore further investigated the pH-dependence of Kir2.2[G178D] sub-state behavior. While the single conductance of WT Kir2.2 was independent of pH (Fig. 2B), the sub-state occupancy was strikingly pH-dependent in G178D channels (Fig. 2A). The same four conducting states were detectable at each pH (Fig. 2A, C) with pH-independent sub-state amplitudes (Supplementary Table 2, Fig. 2D), but there was a clear pH dependence to the sub-state occupancy, with a gradual shift from the lowest-conductance sub-state ($O_1$) being predominant at pH 6.2, to the highest conductance ($O_4$) being almost exclusively present at pH 8 (Fig. 2C). Such dependence on pH supports independence of the full closing events, and points to a role of the protonation state of the

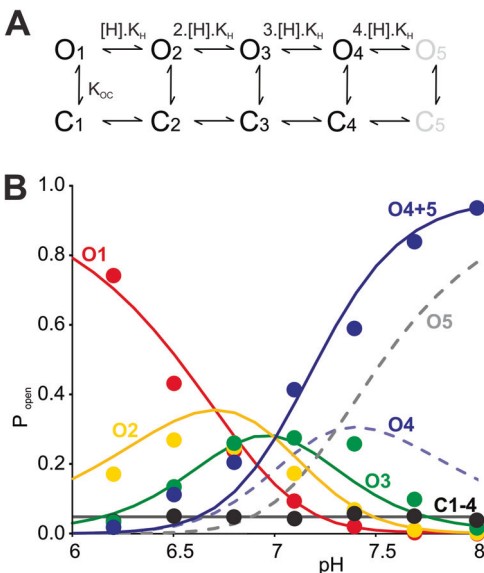

**Fig. 3 | Kinetic model of Kir2.2[G178D] sub-state gating. A** A scheme representing 5 open states (O1 to O5), and 5 brief closed states (C1 to C5). Sequential transitions from state 0 up to state 5 result from single subunit protonation steps. **B** Experimental data (dots) and model prediction, assuming $K_H = 2 \times 10^{-7}$ M$^{-1}$, $K_{OC} = 5 \times 10^{-2}$.

introduced G178D residues in occurrence and conductance of the sub-states.

Over the negative membrane potentials at which each conductance state was experimentally resolvable (from −160 to −60 mV) the subconductance states were clearly voltage-independent, with all sub-state conductances scaling linearly with transmembrane potential (Fig. 2E). In excised patches with one or a few Kir2 channels, channel activity rapidly decays at positive membrane potentials (i.e., with outward currents), both for WT Kir2.2 and Kir2.2[G178D], potentially due to the well-known, and almost unavoidable voltage-dependent blocking effect of residual intracellular polyamines[2,24–26]. Recordings from patches with one or a few WT and G178D mutant channels, obtained at a relatively low positive potentials before complete run-down, demonstrated qualitatively the same behavior as at corresponding negative membrane potential in each case (Supplementary Fig. 1), i.e., no sub-states in the WT and multiple (albeit poorly resolved) sub-states in G178D mutant channels.

## Sub-state gating can be modeled as non-cooperative subunit protonation events

Given that most sub-state conductance transitions occurred between adjacent sub- states, we explored a simple kinetic model to explain G178D intra-burst sub-state gating (Fig. 3A). The model assumes that each subunit within the tetramer can be independently protonated, with channel conductance decreasing for each successive protonation. Since each subunit can also undergo independent closed-open transitions, the model implies five conductive states (O$_1$ to O$_5$), and five "fast" intra-burst closed states (C$_1$ to C$_5$). With the assumption that there is no cooperativity in the protonation events, the model is controlled by only two independent parameters, the proton dissociation constant K$_H$ and the intra-burst closed equilibrium constant K$_{OC}$. With the chosen equilibrium constants, the model predicts the experimentally observed C and O$_1$-O$_3$ sub-state probabilities over the range of pH 6–8 (Fig. 3B). Experimentally, we only resolve four conductance states, and if we assume to have captured the two highest sub-conducting levels O$_4$ and O$_5$ (Fig. 3B, dashed lines) combined as the highest O$_4$ conductance level in experiment, the sum of O$_4$ and O$_5$ (Fig. 3B, purple solid line) gives excellent prediction of the overall behavior.

## Sub-state gating can also be introduced into Kir2.1 channels

To probe the relevance of the introduced ionizable HBC residue more broadly, we introduced an acidic residue at the equivalent position in the archetypal strong inward rectifier Kir2.1 channel. Overall, this Kir2.1[G177E] mutant channel demonstrated very similar behavior to the Kir2.2[G178D] or [G178E] mutations: At neutral pH, the Kir2.1[G177E] mutant also demonstrated increased peak unitary conductance (~50pS for Kir2.1[G177E] vs ~ 30pS for WT Kir2.1) and at least 4 conducting states (Fig. 4A, B) that were not present in WT Kir2.1 (Fig. 4C). Similar to Kir2.2[G178D], the distribution of sub-state conductance level occupancies was pH-dependent, with higher conducting levels being favored at higher pH (Fig. 4A). Unlike Kir2.2[G178D], the full conductance of Kir2.1[G177E] increased significantly with pH (Fig. 4A, D); however, all sub-state conductances scaled proportionally with pH (Fig. 4D, see inset). Thus, mutation of a conserved glycine at the HBC to an aspartic/glutamic acid in both Kir2.1 and Kir2.2 results in (a) increased unitary channel conductance, (b) occurrence of very similar sub-conductive states, and (c) a very similar pH-dependence of conducting sub-state occupancies.

## The number of introduced G177E mutations in Kir2.1 and the number of sub-states are not trivially related

Thus far, the sub-state behavior could trivially be a consequence of independent protonatable groups. Since at least three distinct conductance levels have been resolved experimentally, the most obvious candidates causing this phenomenon are the four introduced aspartate residues themselves. To test this directly, we generated several tandem dimers of Kir2.1 with either two or only one G177E mutation per dimer, further referred to as **E-E** or g-**E** dimers, respectively. Both of these constructs formed functional channels at neutral pH when expressed in COSm6 cells (Fig. 5C, E). Expressed **E-E** dimers resulted in **E-E E-E** channels, with four mutated residues per channel, as in the parental Kir2.1[G178E] construct. **E-E E-E** channels were functionally essentially identical to tetrameric Kir2.1[G177E] channels at each pH (Fig. 5A, C), with four resolvable open states of similar amplitudes and pH-dependence (Fig. 5B, D). Surprisingly, g-**E** g-**E** channels, also still clearly demonstrated four conducting states (Fig. 5E), although with a somewhat different distribution of amplitudes and occupancies. This unpredicted finding, i.e., that g-**E**, containing only two introduced G177E residues per channel, still generates four distinct conducting sub-states, indicates a more complex mechanism of sub-state generation, potentially involving additional ionizable groups.

## Additional ionizable residues contribute to sub-state generation

As discussed above, for protonation of the introduced acidic residues to be involved in pH-dependent sub-states, the effective pKa must be shifted to a much higher pH than that of solvent-exposed aspartate or glutamate side chains. To estimate pKa values for these aspartates and other potentially ionizable residues in the pore, we used a conductive snapshot of the Kir2.2[G178D] protein from our previous simulations (PDB ID: 6M84)[12] and uploaded it to a webserver which predicts the protonation state of ionizable residues at different pH values[27]. Within the inner cavity, not only the G178D side-chains, but also the wild type D173 residue side-chains[28,29] showed predicted pKa values close to the pH 6–8 range over which sub-state titration occurs (Supplementary Table 3). This suggests that the titration of both residues G178D and D173 may contribute to sub-state generation.

To examine the contribution of the protonation state of the naturally occurring aspartate residues in the inner cavity, we neutralized these residues by introducing the additional [D172N] mutation into the g-**E** dimer background to obtain the ng-n**E** dimer construct. The ng-n**E** ng-n**E** channel thus contains only two ionizable Kir2.1[G177E] residues within the channel inner cavity. The ng-n**E** construct was again functional in excised patches, with peak unitary conductance of ~47 pS at pH 7.4. Thus, although unitary conductance

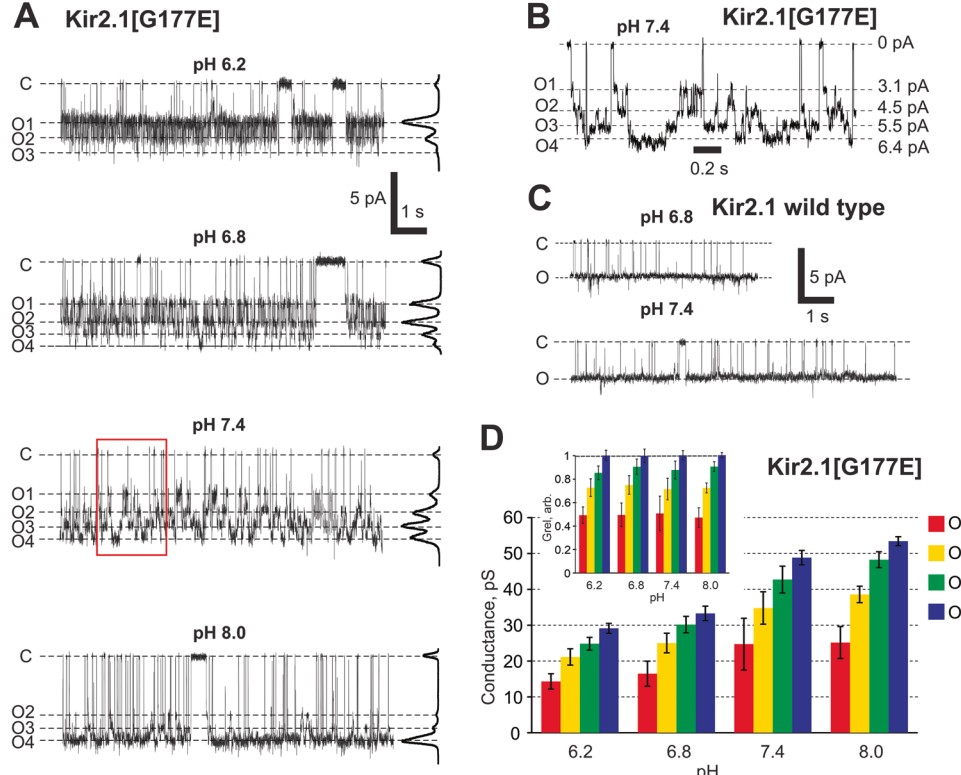

**Fig. 4 | pH-dependent sub-state conductance in in hKir2.1[G177E].**
**A** Representative traces of Kir2.1[G177E] channel activity at different pH. All-point histograms are presented on the right of each trace. **B** Expanded interval of the trace from (A) reveals a closed and four conducting states at pH 7.4. **C** Representative traces of WT Kir2.1 channel activity reveal only a single open state at pH 6.8 or 7.4. **D** Sub-states amplitudes of cKir2.1[G177E] mutant at −120 mV

membrane at different pH. *Inset:* normalized conductances of individual hKir2.1[G177E] mutant sub-states at different pH. All measurements were made on single channels in symmetric 150 mM KCl buffer with indicated pH. Data in **D** shown as mean ± s.e.m. ($n = 100$–$1000$ transitions within each patch, $N = 2$ patches, for each conductance level).

is again significantly higher than that of WT Kir2.1 (~30 pS), it is similar to both the Kir2.1-G177E (~50 pS) and the parental g-**E** dimeric constructs (~48 pS). In contrast to the other mutants, ng-n**E** now clearly demonstrated only three well-resolved conductive sub-states, with distinct amplitude distribution (Fig. 5G), while still following a similar pH dependence (Fig. 5H). These three conductances can then be simply explained as the result of zero, one, or both G177E residues being protonated. This indicates that the ionizable D172 residue must contribute to the higher number of sub-state conductances that are seen in the g-**E** and other constructs.

**Negative charges in the cavity determine conductance**
The above data indicate that the sub-state transitions represent individual protonation/deprotonation events at residues Kir2.2[G178D] and D173 in the inner cavity. This raises a question as to how such protonation events manifest as significant gating events – i.e., degrees of 'opening' of the channel. To investigate the consequences of such discrete protonation events on channel conductance, we employed MD simulations. Building on the simulations from our previous work[12], we started with a snapshot of an open and conductive conformation of the Kir2.2[G178D] protein (again with the mutation K62W included in the structure to substitute for the necessity of anionic phospholipids to activate Kir2 channels[9,12]). To mimic the pH-range of our patch-clamp experiments, we generated four separate MD system setups, summarized in Table 1. In the first system, none of the D173 or G178D residues were protonated, i.e., each side chain was negatively charged, resulting in a system (termed $Q_{cav}$ −8) with a net charge of −8 in the central cavity. Assuming that individual residues would be protonated in a stepwise manner with decreasing pH (corresponding to

acidification of the medium), we protonated one each of the four side chains of D173 and G178D (i.e. both in subunit A) in the second ($Q_{cav}$ −6) system. In the third system ($Q_{cav}$ −4), two diagonally opposed D173 and G178D (i.e. in subunits A and C) each were protonated. We also introduced a WT-$Q_{cav}$ −2 system, in which residue 178 was the wild type G178 (in all 4 subunits), and only two opposing D173 (i.e. in subunits A and C) were protonated.

As in the electrophysiological experiments, $Q_{cav}$ −8 (mimicking conditions at high pH) was the most conductive system with an average of 12.7 (S.D. 1.2) $K^+$ ions being conducted in 1 μs. As shown in Table 1 and Fig. 6, the conductance decreased as D173 and G178D were increasingly protonated: with an average of 10.7 (S.D. 3.5) $K^+$ ions per microsecond in system $Q_{cav}$ −6, and only 1.0 (S.D. 1.0) $K^+$ ion per microsecond in the $Q_{cav}$ −4 system. Qualitatively, the large drop in conductance between $Q_{cav}$ −6 and $Q_{cav}$ −4 (compared to the minor change of conductance between $Q_{cav}$ −8 and $Q_{cav}$ −6) might correspond to the unevenly spaced subconductance levels seen in experiment, and support the possibility that there could be multiple additional small and experimentally unresolved steps between the full conductance and the next clearly resolved conductance. As in our previous study[12], the conductance was lower in the WT-$Q_{cav}$ −2 system, potentially reflecting the experimentally lower conductance of WT Kir2.2.

The parallel findings that conductance decreases at lower pH in experiment, and as protonation state increases and thereby net negative charge decreases in MD simulations, argue that protonation of ionizable residues in the central cavity is the controlling molecular event. As the next step, we analyzed how such changes determine the different levels of conductance.

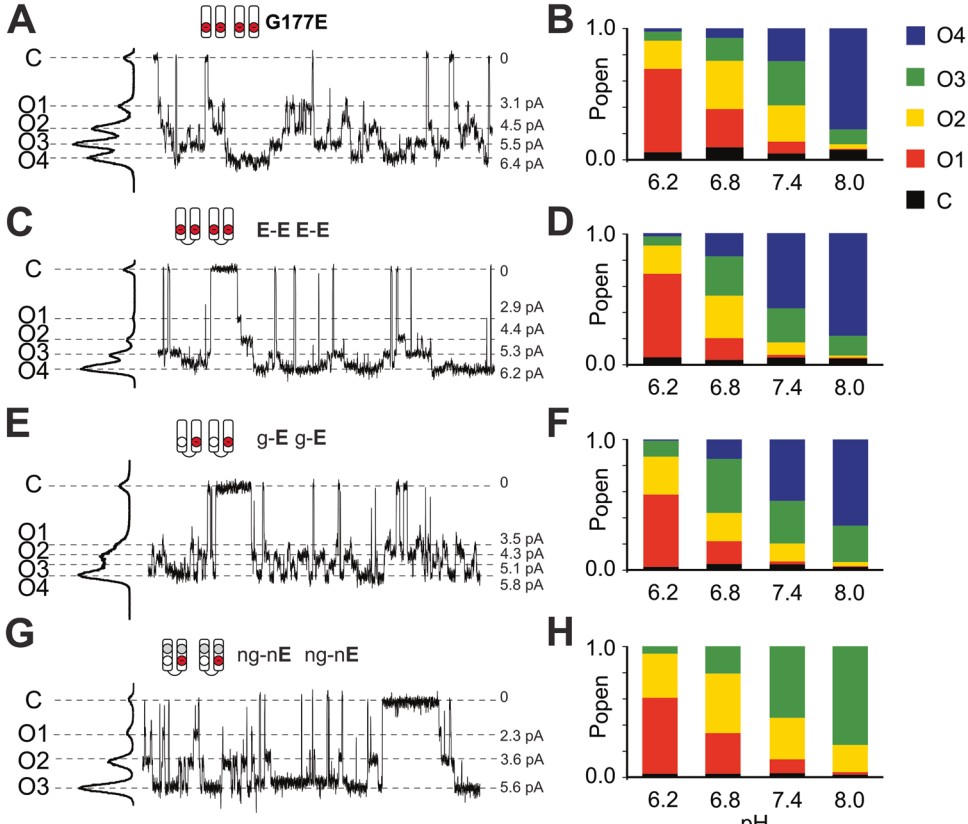

**Fig. 5 | Conductive states of hKir2.2[G177E] and mutant tandem constructs at different pH levels. A, C, E, G** Representative traces of monomeric Kir2.1[G177E] (A) or dimeric constructs (see text) at pH7.4. Amplitude histograms to the side of the trace indicate subconductance levels. Pictograms represent numbers of introduced ionizable residues at residues in the cavity pore per each two subunits of the functional tetrameric channel. **B, D, F, H** Distribution of corresponding sub-state probabilities at different pH levels.

## Protonation leads to changes in K+ ion occupancy, pore solvation, and gate diameters

The flux of single K+ ions contributing to the conductance rates summarized in Table 1 is visualized in Fig. 6 (top panels), Supplementary Fig. 2, and Supplementary Movies 1 and 2. As is clearly seen, the number of negatively charged residues in the pore essentially determines the K+ occupancy in the pore. Between the selectivity filter (SF) and M308 of the G-loop, the pore pools an average of 9.3 K+ ions in the $Q_{cav}$ −8 system, while K+ occupancy steadily decreases to 6.3 (in $Q_{cav}$ −6) to 5.6 (in $Q_{cav}$ −4), as protonation increases (Supplementary Fig. 3).

### Table 1 | Overview of simulations and K+ conductance:

|  | $Q_{cav}$ −8 | $Q_{cav}$ −6 | $Q_{cav}$ −4 | Control |
|---|---|---|---|---|
| Residue protonation |  |  |  |  |
| D173 charge | −4 (-\|-\|-\|-) | −3 (H\|-\|-\|-) | −2 (-\|H\|-\|H) | −2 (-\|H\|-\|H) |
| G178D charge | −4 (-\|-\|-\|-) | −3 (-\|-\|H\|-) | −2 (H\|-\|H\|-) | 0 (G\|G\|G\|G) |
| SF conduction events / 1 µs |  |  |  |  |
| md1 | 14 | 14 | 0 | 1 |
| md2 | 12 | 7 | 1 | 2 |
| md3 | 12 | 11 | 2 | 2 |
| Avg. | 12.7 | 10.7 | 1.0 | 1.7 |

The protonation of ionizable residues in the central cavity and the number of conduction events per µs are shown for each simulation system. Thereby, individual residues of the protein homotetramer can be protonated (H), deprotonated (-), or WT G178 (G). For each system, three 1 µs runs were simulated (md1-3). Conduction events were counted when a K+ ion left the SF S1 site into the extracellular solvent.

The lack of any G178D aspartates in the WT-$Q_{cav}$ −2 system further reduces the occupancy to 3.6 K+ ions.

As previously demonstrated, the introduction of negative charges at residue 178 in the bundle crossing region increases the diameter at the HBC gate[12]. Thus, the position of the hydrophobic isoleucine residue 177 within the Kir2.2 HBC (Fig. 1A), is strongly affected by changes of the protonation state of both G178D and D173 (Fig. 7). As protonation increases and the net charge drops within the cavity, the pore diameter at this location narrows, on average, from 8.2 Å (in $Q_{cav}$ −8) to 6.1 Å (in $Q_{cav}$ −4). Similarly, although M181 side chain flexibility is greater than that of the hydrophobic I177, increasing protonation also results in a narrowing of the most frequently sampled M181 minimum distances from 10.6 Å ($Q_{cav}$ −8) to 8.9 Å ($Q_{cav}$ −4), with an additional peak at 6.2 Å (Fig. 7). Minimum distances in the HBC gate of the WT-$Q_{cav}$ −2 simulation are similar to values of the $Q_{cav}$ −4 system. Additional pore-constricting residues in the inward direction are M308 and M302 of the G-loop. Interestingly, the presence of the G178D mutant, whether protonated or not, has a pronounced effect on M308, leading to widening in the mutant channel compared to WT-$Q_{cav}$ −2 simulations. The most frequent minimum distances range from 9.4 Å ($Q_{cav}$ −8) to 8.6 Å ($Q_{cav}$ −4), but much narrower M308 residues (≤5.4 Å) in a considerable fraction of WT-$Q_{cav}$ −2 simulations. M302 is less influenced by the G178D mutation, constricting the G-loop to minimum distances around ~6.0 Å in all systems. The constriction of these gates (Fig. 7) is accompanied by temporarily desolvated periods (Fig. 6 and Supplementary Fig. 2, bottom panels). Thus, in addition to direct effects on K+ occupancy, increasing protonation also decreases pore solvation in the HBC and the G-loop gate as the cavity charge decreases, resulting in increasing interruption of the continuous

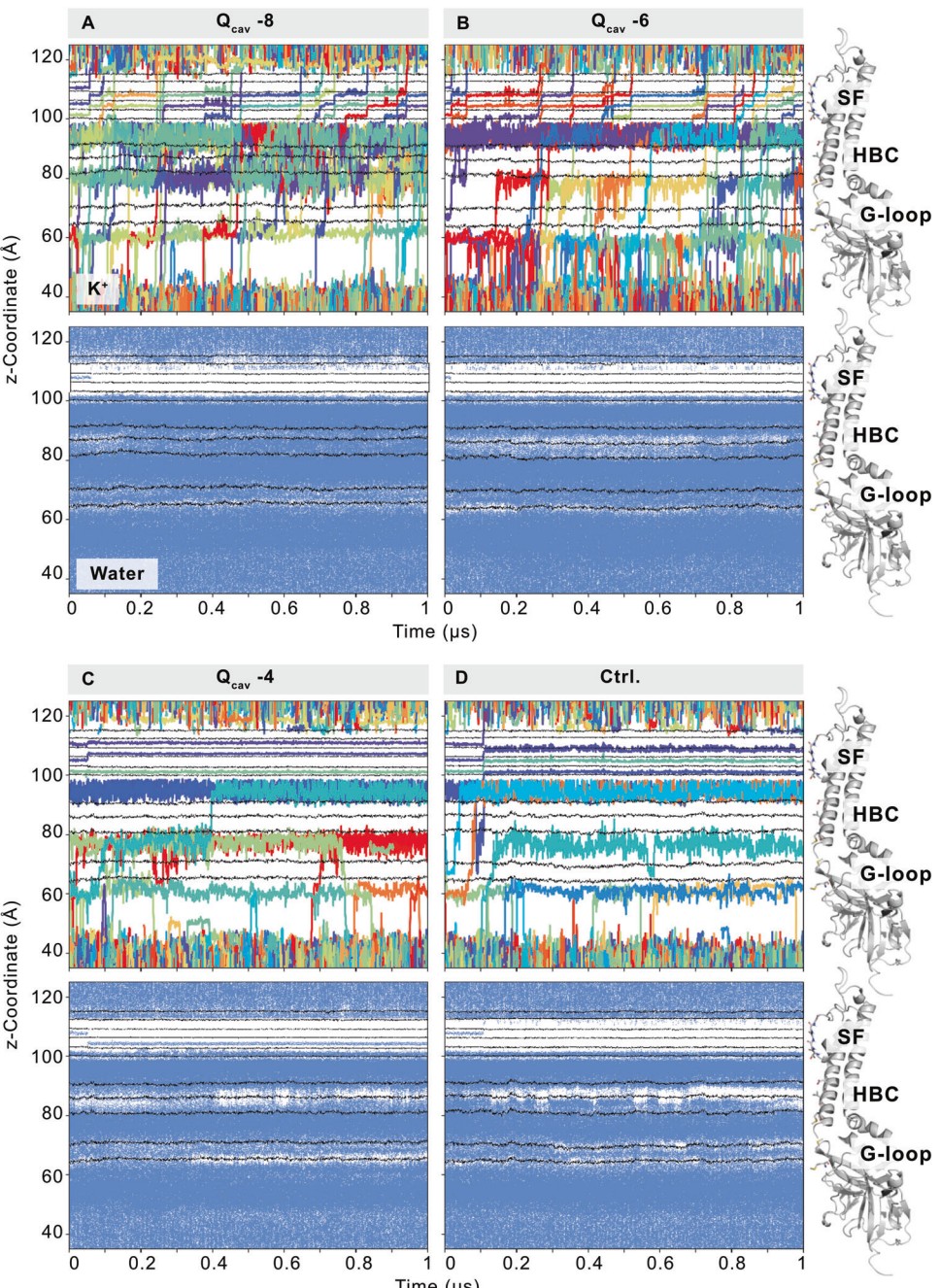

**Fig. 6 | Deprotonated residues in the central cavity increase ion flux and pore solvation.** Top panels (**A**, **B**) illustrate the flux of individual K$^+$ ions through the channel pore (z-coordinate of the simulation box). Single conduction events can be compared between systems with varying protonation of residues in the upper pore (D173 and G178D). In all simulations, an electric field was applied in the outward direction. The bottom panels (**C**, **D**) show the corresponding pore solvation, where blue dots represent the occurrence of water molecules in the pore (see the Methods). Black lines in the plots correspond to the center-of-mass of residues in the selectivity filter (SF backbone-O: G147, Y146, G145, I144, T143; SF side-chain O: T143), the wild type aspartate (D173), and gating residues (HBC: I177, M181; G-loop: M308, M302). On the right side, these residues are shown in stick representation on a single protein subunit. Only md1 of each simulation system is shown, while plots of all replicas can be found in Supplementary Fig. 3.

solvation pathway of the pore. Therefore, de-wetting of the solvation pathway and the presence of the uncharged glycine at residue 178 in the WT-Q$_{cav}$ −2 system decreases the probability of an ion traversing the pore.

## Discussion
### Channel gating and sub-states
Since the first recordings of recognizably individual ion channels, it has been evident that most ion channels exhibit stereotypically on-off behavior, with two experimentally measurable current levels, one that is indistinguishable from zero and one that is characteristic of the specific channel conductance under a given ionic condition. This behavior is described as the channel existing in two states, one 'closed' and one 'open', and conceptually envisioned as resulting from there being a 'gate' in the permeation pathway that allows ion flow past it or not, when either open or closed. Crystal and cryo-EM structures have provided direct structural correlates, with some channel structures clearly showing tight steric or hydrophobic constrictions through

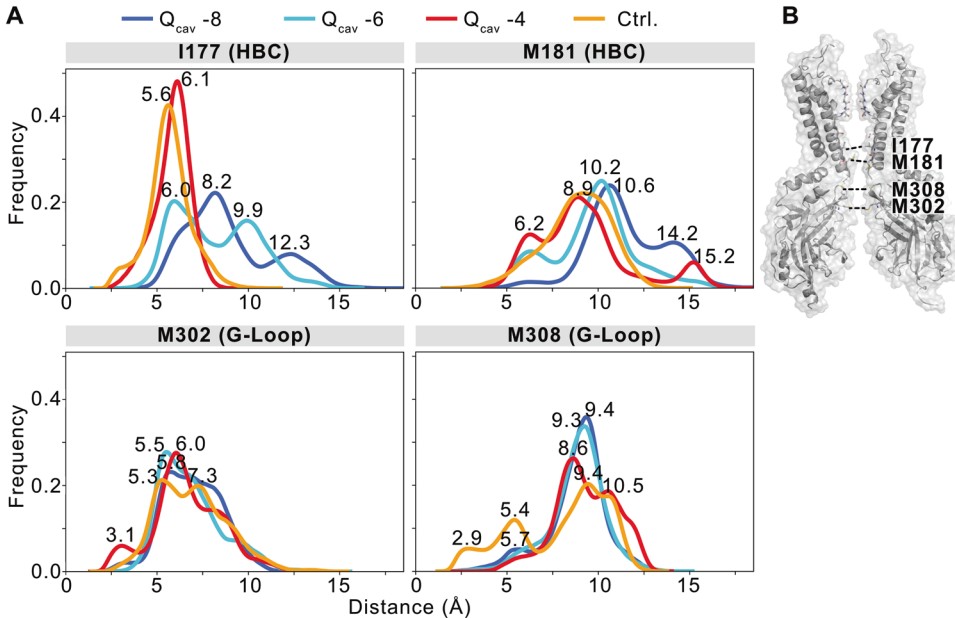

**Fig. 7 | Negative charges in the pore increase the gate diameter, especially of the hydrophobic I177 in the HBC gate. A** Histograms show the distribution of gate diameters for residues in the HBC gate (I177, M181) and the G-loop gate (M302). Different colors correspond to simulation systems with varying protonation of D173 and G178D. **B** The location of constricting residues in the pore is shown on two opposing protein chains of the cKir2.2 tetramer.

which ion permeation could not occur, and that are relieved by conformational changes under 'activating' conditions, although often the underlying structural changes seem to be quite subtle. This is the case for the cation channel superfamily in general, including Kir channels, which open in response to channel-specific stimuli, including water-soluble or membrane-soluble ligands[1,7,9,30]. Many ligands interact with the regulatory cytoplasmic domain (CTD) and control opening through physical coupling to the 'gate', a steric restriction that can be too narrow to allow ion flow, located at the M2 helix bundle crossing (HBC) at the bottom of the transmembrane pore domain (TMD). Multiple high-resolution structures of different eukaryotic Kir channel family members have been obtained, although, except for one $K_{ATP}$ (Kir6.2) structure, these have generally been in a closed, likely non-conductive state[5,9–11,31,32]. Prokaryotic KirBac3.1[S129R] and Kir2.2[G178D][12] (or G178R) mutations essentially lock the channel open functionally, resulting in crystal structures that show a minor expansion at the HBC. When placed in a membrane and solvated in silico, these Kir2.2[G178D] and G178R structures rapidly undergo further expansion at the HBC, enabling hydration of the HBC and $K^+$ ion flux across the membrane[12,30].

A complication to the simple notion of channels having a single 'gate' is that most are multimeric, and often homomeric, in which case they should have as many 'gates' as there are subunits. In the case of voltage-gated K channels, there are clearly four independent voltage sensors and, in the case of the Kir channels, there are always four binding sites for any allosteric ligands[1]. A classic explanation of single step 'opening' is that the step from zero to full conductance is the last and unified step that follows hidden conformational changes resulting from the gating ligands or sensors, and kinetic analyses have provided some very compelling kinetic models that fit this paradigm[16,33–36]. But there is still a lingering discomfort with this framework, since a final single opening step cannot easily be accomplished in multimeric (e.g., tetrameric Kir) channels. One notion is that channel opening actually occurs in multiple steps, but that these are so highly cooperative that each step cannot be experimentally resolved. Another possibility is that there is cooperativity in the effects of each step on conductance, which becomes maximal with the first subunit 'opening', or which is not measurable until all subunits are 'open'. For these reasons, the

appearance of very consistent sub-states in the Kir2.2[G178D] mutant channels becomes a particularly intriguing model system in which to investigate the contribution of individual subunits to channel gating.

## The atomistic basis of sub-states

There have been previous reports of sporadic subconductance states in wild type Kir2.1[15] or heteromeric Kir2.1/Kir2.2 channels[14], as well as subconductance states induced by blocking ions or by introduction of unnatural amino acids in the SF[37]. However, we have not consistently observed more than one open conductance state for either of the wild type Kir2.1 or Kir2.2 channels, regardless of pH. The single-channel sub-state behavior of G178D mutant currents is clearly resolvable with sub-state lifetimes of ~$10^{-3}$ s, and it is very tempting to suggest that each state represents a distinct subunit conformation that changes the steric limitation on ion accessibility. Given that the sub-state occupancy is titratable by pH, it is clear that protonation steps are involved and, by limiting the number of subunits with titratable acidic side-chains, we have succeeded in experimentally limiting the number of sub-states. One question that then arises is whether it is even conceivable that single protonation steps could underlie such relatively long lifetimes? The pKa for free aspartate side chains in solution is ~3, and assuming that the protonation rate is a diffusion-limited association ($k_{on}$ ~ $10^{10}$ $M^{-1}s^{-1}$), the effective on-rate (and off-rate at pH 3) would therefore be immeasurably fast (~$10^7 s^{-1}$). The apparent pKa for sub-state gating is ~7 (Fig. 3). An upshift of the aspartate pKa from 3 to 7 within the constricted space that could prevail within the inner cavity would therefore slow on- and off-rates by $10^4$ (i.e., ~10,000-fold), bringing them both into the millisecond range at neutral pH. Thus, the striking conclusion that the distinct sub-states are simply a direct consequence of the protonation state of the G178D side chains is quite reasonable.

## Ionization of side chains in the pore controls conductance

One obvious and straightforward conclusion arising from our current and previous[12] analyses is that increase of the net negative charge in the pore, at least at the narrow region near the HBC, leads to higher $K^+$ currents through it. Higher up in the pore, such electrostatic effects on $K^+$ conduction may be weaker; single channel conductance of

Kir2.1[D172N, or Q] may be slightly lower than wild type, but not markedly so[28,38], although recent MD simulations suggest that conductance is inhibited if all four Kir2.2 D173 residues are ionized at lower membrane potentials[30]. On the other hand, introduction of polyamine block by introduction of negative charges at essentially any cavity-lining position, up to the entrance of the SF, in an otherwise neutral Kir6.2 inner cavity, introduces similarly strong polyamine blocking properties[26].

Our simulations suggest that the mechanism of pore negativity-driven increases in conductance primarily relies on increased overall $K^+$ occupancy of the pore, but also on consequent pore widening and increased solvation. As our previous work suggests[12], placement of a ring of negative charges at the Kir2.2 HBC (G178D) results in quite minor (~1 Å) lateral expansion of the M2 helix backbone (at G178D), with the charged side chains not facing the pore lumen but instead facing adjacent pore-lining helices. Our previous MD simulations suggest that this local re-arrangement is sufficient to slightly increase wetting at the HBC constriction. By attributing fixed protonation states to the two key pore-lining residues (Kir2.2 mutant G178D and wild type D173), with predicted pKa values closest to cytoplasmic pH (Supplementary Table 3), we could model $K^+$ transport through the channel under conditions mimicking different intracellular pH. In four systems containing various net negative charges located within the inner cavity, we observed gradual expansion of the HBC region with increasing net negative charge. Expansion was significant at the hydrophobic I177 residue (Fig. 7), resulting in higher $K^+$ occupancies at the HBC and in the inner cavity (Supplementary Fig. 3).

The recognition that titration of both G177E and D172 residues contributes to the sub-state behavior in the highly homologous and archetypal Kir2.1 channel, shows that this sub-state gating is a generalizable phenomenon, dependent on the presence of titratable residues in the inner cavity. Clearly, the contribution of each residue is not equal; this is most evidenced by there being still (at least) 4 distinct sub-states in g-**E** channels (with 4 D172 and 2 G177E residues), but only three very distinct sub-states in ng-n**E** (with 4 D172N and only 2 G177E titratable residues). One possibility to reconcile these seemingly disparate behaviors is that the major contribution comes from the G177E residues, and that the contribution of D173 residues is minimal, and may depend on the presence or absence of G177E residues. Thus, four levels in **E-E E-E** channels would be due to 0, 1, 2, or 3 (or 4) G177E residues being protonated, with the conductance difference between 3 and 4 protonated G177E side chains being too small to measure experimentally. The three conductance levels in ng-n**E** ng-n**E** channels would then be due to 0, 1, or both G177E residues being protonated, and additional sub-states would emerge in the g-**E** g-**E** channels as D172 residues are additionally titrated.

In conclusion, we report striking sub-state gating behavior as a result of introduction of acidic residues at Kir2.2 residue 178 or residue 177 in Kir2.1 within the HBC region. The results can be explained by conductance through the channel being directly dependent on the protonation state of individual G178D residues, with each additional ionized side chain causing a marked stepwise increase in conductance as a result of increased wetting and increased $K^+$ occupancy in the inner cavity. The results provide a dramatic illustration of channel sub-state 'gating', resulting simply as a consequence of changing charge states within the inner cavity that alter ion occupancy and hence conductance.

## Methods
### Molecular biology
Point mutations in both chicken pCDNA3.1-Kir2.2 and human pCDNA3.1-Kir2.1 were introduced using the QuickChange II method (Agilent Technologies) with the entire coding region verified by sequencing. Tandem dimers of Kir2.1 were made from two identical full-length protein DNA sequences, consecutively (C-terminus to N-terminus) connected with a flexible GENLYFQGQGSG linker.

### Electrophysiology
CosM6 cells (originally derived from monkey kidneys) were transfected with 0.3-1 µg of pCDNA3.1-Kir2.2 and pCDNA3.1-Kir2.1 constructs with an addition of 0.4 µg of pcDNA3.1-GFP per 35 mm Petri dish using FuGENE6 (Promega). The cells were used for patching within 12–48 h after transfection. For patch-clamp experiments, symmetrical internal potassium buffers ($K_{int}$) were used: 148 mM KCl, 1 mM EGTA, 1 mM $K_2$EDTA, 10 mM HEPES (pH 6.8, 7.1, 7.4, 7.7) or 10 mM MES (pH 6.2, 6.5) or MOPS (pH 8.0, 8.6). Data were acquired at 3 kHz, low-pass filtered at 1 kHz with Axopatch 1D patch-clamp amplifier and digitized with Digidata 1320 digitizer (Molecular Devices). Data analysis was performed using the pClamp software suite (Molecular Devices). Pipettes with resistance of ~ 2–8 MOhm in symmetric $K_{int}$ were pulled from Kimble Chase 2502 soda lime glass with a Sutter P-86 puller (Sutter Instruments). All measurements were carried out on excised inside-out patches at −120 mV membrane potential, or as specified in the text.

### Molecular dynamics simulations
Gromacs 2021.5[39] was used to perform MD simulations and analyses. Simulation systems were based on the setup of our preceding paper[12] using a frame from an open, conductive channel (md1, $t = 50$ ns) as a starting point. In brief, the cKir2.2 protein in complex with short-chain PI(4,5)$P_2$ (PDB ID: 6M84) was embedded in a bilayer membrane consisting of palmitoyl-oleoyl phosphatidylcholine (POPC) lipids, and solvated with water and 0.2 M KCl. As in our previous work, we used the amber99sb force field for the protein[40], Berger lipid parameters for the POPC lipids, the SPCE water model, and corrected monovalent Lennard-Jones parameters for ions[41]. Short-chain PIP$_2$ parameters were taken from our previous work. 1.0 nm cut-offs were set for short-range Coulomb and van der Waals interactions, while long-range electrostatics were treated with the particle-mesh Ewald method[42]. Bonds were restrained with the LINCS algorithm[43]. The production runs were simulated in an NPT ensemble. We used the velocity-rescale thermostat to couple three independent temperature coupling groups (protein, lipids, and solvent) to a temperature bath of 310 K with $\tau = 0.1$ ps. The pressure was kept constant at 1 bar with the Parrinello−Rahman barostat ($\tau = 2$ ps)[44].

pKa values were predicted with the PlayMolecule ProteinPrepare webserver, which uses *PROPKA 3.1*[45] and *PDB2PQR 2.1*[46], to guide the protonation of ionizable residues in the systems. We used a conductive snapshot of the Kir2.2[G178D] protein from our previous simulations[12] with standard values. After assigning protonation states for the residues D173 and G178D with Gromacs, the systems were neutralized by removing excess ions. Subsequently, the energy of each system was minimized with the steepest descent algorithm. To adapt the solvent in the pore to the new charge environment, we first performed a short 100 ps NVT equilibration with strong position restraints on all heavy atoms with a force constant (fc) of 100,000 kJmol$^{-1}$nm$^{-2}$, followed by a 40 ns NPT equilibration with weaker restraints (fc = 1000). For the production simulations, the Y146 side chain atoms were restrained with positional restraints (fc = 1000) as in our previous work[12]. An electric field of 40 mV·nm$^{-1}$ (which converts to ~580 mV in total over the simulation box) was applied in the outward direction. G178D was mutated back to G (G178D(G)) with the Swiss PDB Viewer[47] in our control simulation.

### Analysis of MD trajectories
**Ion flux and pore solvation.** Trajectories were aligned along the selectivity filter backbone (residues 143–148). For the analysis, a frame every 1,000 ps was written out to be processed for the following analysis, resulting in 1001 frame per run. Positions along the pore axis (the z-coordinate) were written out for $K^+$ ions and water molecules in

the pore. z-Coordinates-over-time were written out and filtered with an in-house script and plotted with Matplotlib. Therefore, selection cylinders were used: The selection cylinders had a height of 11 nm, a radii of 2.5 nm for $K^+$ or 1 nm for water, and were centered on the center-of-mass (COM) of M181 residues. For the selectivity filter, another, smaller cylinder was used, ranging between the backbone-oxygen COMs of T143-Y146, and a radius of 0.25 nm.

**$K^+$ occupancy in the pore.** Z-coordinates of $K^+$ ions in the pore (visualized in Fig. 6 and Supplementary Fig. 2 over time) were used to calculate histograms (0.5 Å bins). For each system, 1001 frames from each of the three replicas were used.

**Minimum distances.** were measured between opposing protein sub-units for all residues, thus resulting in two distance pairs (chains 1–3 and 2–4). The corresponding distance-versus-time plots can be found in Supplementary Fig. 4.

### Reporting summary
Further information on research design is available in the Nature Portfolio Reporting Summary linked to this article.

## Data availability
The data that support this study are available from the corresponding authors upon request. MD simulation data and scripts used to produce figures for this study have been deposited in a Zenodo repository [https://doi.org/10.5281/zenodo.8059438]. Electrophysiological data have also been uploaded to Zenodo [https://zenodo.org/record/8071768].

## Code availability
Computer code produced in this study, including scripts for the ion flux and pore solvation MD analyses are available upon request, and have been deposited in a Zenodo repository [https://doi.org/10.5281/zenodo.8059438].

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

## Acknowledgements
This work was supported by NIH R35 HL140024 to C.G.N., NIH R03 TR003670 to S.J-.L., Austrian Science Fund grant nr. W1232 (Moltag) to M.B.-J. and A.S.-W, and the post-doc program "Zukunftskolleg" ZK-81B to E.-M.Z.-P. Molecular Dynamics simulations were achieved with resources provided by the Vienna Scientific Cluster (VSC) and the Texas Advanced Computing Center (TACC).

## Author contributions
G.M., S.-J.L., and C.G.N. conceived the project. G.M. and M.B.-J. carried out electrophysiological experiments. G.M., M.B.-J., and C.G.N. analyzed the experimental data. M.B.J. performed MD simulations, M.B.-J., E.-M.Z.-P., and A.S.-W. analyzed the MD data. S.J.L. and M.B.-J. carried out mutagenesis and molecular biology manipulations. G.M., M.B.-J. and C.G.N. wrote the manuscript, which was edited by the other authors.

## Competing interests
The authors declare no competing interests.
