## [Peer Review File · Nature Communications]

Subunit gating resulting from individual protonation events in Kir2 channelsReviewers' Comments:

Reviewer #1:

Remarks to the Author:

Thank you for the invitation to review 'Subunit gating resulting from individual protonation events in Kir2 channels' by Makshev and colleagues. This is a very interesting study that describes the effects of protonation of substituted side chains on the conductance of Kir2.1 or Kir2.2 channels. This study demonstrates that in Kir2.1 or Kir2.2 channels mutated to have a protonatable side chain (Asp or Glu) in the helix bundle crossing, multiple subconductance states arise, likely due to protonation of the substituted Asp or Glu side chain. The study includes additional deeper analysis of this finding, demonstrating that the number and level of subconductance states is likely influenced by other nearby residues (D172 in Kir2.1 or D173 in Kir2.2), and simulating the effects of protonation on K⁺ occupancy/permeation and pore diameter to develop some mechanistic explanations of their experimental findings. A strength of this paper is that it provides direct observation of the pKa shifts that may occur when protonatable acidic side chains are closely oriented in a multimeric protein, and also what is likely a direct observation of the lifetimes of protonated and deprotonated states. While the particular sites studied here are 'engineered' in the sense that they are not protonatable in wild type channels, the findings and observations likely generalize to other nearby sites such as the inner cavity D172 or D173 that controls sensitivity to blockers in Kir channels. The experimental data are of exceptionally high quality and clarity, and the manuscript describes a complex topic very clearly.

Comments:

I have no major concerns about the content of the manuscript, but I have some comments and questions that would strengthen the paper if addressed.

1. The authors' references to the connection between 'gating' and 'conductance' are unintentionally vague. I don't feel that the experiments can distinguish between a change in conductance (arising from localized alterations in things like ion occupancy), versus some independent conformation change that most would associate with a gating motion of a single subunit. Most of the paper and discussion describes what I feel is the most likely and fair explanation for the change in conductance, that being relatively independent protonation of side chains on different numbers of subunit. Therefore, statements like this (line 51, line 95, line 429) could be omitted or altered without affecting the main message. As written, the statements might also be taken to imply that subconductance levels in other channels might arise from non-gating related transitions (such as individual subunit protonation), which also is not supported. This would be a minor change that would help.
2. It is interesting that there are occasional brief closures to a fully closed state (within bursts). I agree that this is likely due to some additional gating region like the selectivity filter, as suggested by the authors. However, an interesting observation that is not described is why these full closures appear to happen more frequently from the more conductive states (O3/4) vs. other more protonated/less conductive states (based on the rates in 1C)?
3. ~Line 387. '...net negative charge of the conductive pore leads to higher K⁺ currents through it (although this may be challenged by recent MD simulations suggesting that conductance is inhibited if all four Kir2.2 D173 residues are ionized at lower membrane potentials [29].' Based on other data in Kir2.1 it seems that the pronounced effects of protonation on conductance likely depend a lot on position, even in the inner cavity. I think that commonly used D172N or D172A mutations in the inner cavity of Kir2.1 do not have large effects on conductance, although they clearly affect the functional outcomes of the mutations (shown here) in the bundle crossing. This isn't meant as a counterpoint to the data that are presented (which convincingly show variable protonation-dependent functional outcomes), but rather to consider that there may be important functional effects on properties other than conductance (such as blocker sensitivity).

4. Line 258. '...we protonated one each of the four side chains of D173 and G178D in the second (Qcav-6) system. Could you clarify whether this was done in the same or different subunits? Or whether it matters?

5. Line 415. '... would be due to 0, 1, 2, or 3 (or 4) G178D residues being protonated'... I think this is a typo and likely means 177E in the context of the sentence.

6. Perhaps consider mentioning work by Cymes and Grosman that illustrates protonation dependent side chains in the pore of nAChR channels, or perhaps other scenarios where acidic side chains undergo prominent pKa shifts.

Reviewer #2:

Remarks to the Author:

The authors studied Kir2.2 and Kir2.1 mutations at the inner helix bundle crossing which are known to open the inner gate and to have increased single channel conductances (SCC). These mutants (G178D and G178E in Kir2.2 and the homologous mutation in Kir2.1) displayed rather unusually stable subconductance levels, in comparison to those that are naturally occurring. The four subconductances that the authors observed for the mutants result from independent protonation of the acidic aspartates/glutamates in each channel subunit. The protonation changes the K⁺ occupancy and the pore solvation at the helix bundle crossing and alters the G-gate. Noteworthy, while the SCC increases for WT Kir2.2 from 46 pS to 60 pS for the G178D mutant, the protonation of the mutant can result in a small open state subconductance with x0.45 of the maximal conductance, meaning 27 pS. The SCC which is smaller than that of WT channels, shows that the protonation does not reverse the gating phenotype of gain-of-function for the G178D mutation, but rather causes subconductance levels that are mechanistically different from the changes in gating observed for the G178D/E mutants. Thus, the conclusion in the Abstract and Introduction that gating and conductance are tightly coupled is somewhat correct, albeit limited only to the protonation state of these particular somewhat artificial mutant channels. The study investigates mutants to force the channel into an open state, also resulting in artificial subconductance levels. That the authors can titrate these subconductance levels by protonation and that it is understood how these subconductances can occur, is of biophysical interest, but does not foster our understanding of a naturally occurring gating mechanism, nor can there be any understanding derived for naturally occurring subconductance states of inward-rectifying Kir channels.

Additional points:

-Abstract and discussion: the sentences with the "rectification controller" might need some re-writing sounds too much like lab slang.

-The authors should discuss the limitation of their MDs with using 200 mM K⁺ and applying a voltage of 580 mV.

-At page 7 Kir2.1 is stated to have a SCC of about 30 pS while at page 9 it is stated to have a SCC of about 25 pS.

Response to reviewer comments. Reviewer comments in black. Responses in red.

Reviewer #1

Thank you for the invitation to review 'Subunit gating resulting from individual protonation events in Kir2 channels' by Maksaev and colleagues. This is a very interesting study that describes the effects of protonation of substituted side chains on the conductance of Kir2.1 or Kir2.2 channels. This study demonstrates that in Kir2.1 or Kir2.2 channels mutated to have a protonatable side chain (Asp or Glu) in the helix bundle crossing, multiple subconductance states arise, likely due to protonation of the substituted Asp or Glu side chain. The study includes additional deeper analysis of this finding, demonstrating that the number and level of subconductance states is likely influenced by other nearby residues (D172 in Kir2.1 or D173 in Kir2.2), and simulating the effects of protonation on K⁺ occupancy/permeation and pore diameter to develop some mechanistic explanations of their experimental findings. A strength of this paper is that it provides direct observation of the pK_a shifts that may occur when protonatable acidic side chains are closely oriented in a multimeric protein, and also what is likely a direct observation of the lifetimes of protonated and deprotonated states. While the particular sites studied here are 'engineered' in the sense that they are not protonatable in wild type channels, the findings and observations likely generalize to other nearby sites such as the inner cavity D172 or D173 that controls sensitivity to blockers in Kir channels. The experimental data are of exceptionally high quality and clarity, and the manuscript describes a complex topic very clearly.

We very much appreciate the positive overall comments of the reviewer.

I have no major concerns about the content of the manuscript, but I have some comments and questions that would strengthen the paper if addressed.

1. The authors' references to the connection between 'gating' and 'conductance' are unintentionally vague. I don't feel that the experiments can distinguish between a change in conductance (arising from localized alterations in things like ion occupancy), versus some independent conformation change that most would associate with a gating motion of a single subunit. Most of the paper and discussion describes what I feel is the most likely and fair explanation for the change in conductance, that being relatively independent protonation of side chains on different numbers of subunit. Therefore, statements like this (line 51, line 95, line 429) could be omitted or altered without affecting the main message. As written, the statements might also be taken to imply that subconductance levels in other channels might arise from non-gating related transitions (such as individual subunit protonation), which also is not supported. This would be a minor change that would help.

We accept the reviewers' concern regarding the vagueness of the connection between 'gating' and 'conductance' and so to avoid misleading the audience, we have removed or replaced each of the offending statements.

2. It is interesting that there are occasional brief closures to a fully closed state (within bursts). I agree that this is likely due to some additional gating region like the selectivity filter, as suggested by the authors. However, an interesting observation that is not described is why these full closures appear to happen more frequently from the more conductive states (O3/4) vs. other more protonated/less conductive states (based on the rates in 1C)?

We are not convinced that full closures happen more frequently from the more conductive states. In the example shown in Fig. 1C, at pH 7.4, the rate constants for transitions between sub-states favor the highest level openings, i.e. state occupancy is $O_4 > O_3 > I_2 > O_1$. Rather than consider the absolute number of transitions from each substate to the closed state C, the fraction of transitions to C versus to any state is therefore more relevant. The table shows the calculation for the data in Fig. 1C:

Ox	Ox->C	Ox->O4	Ox->O3	Ox->O2	Ox->O1	Frac Ox>C
O4	163		13	1	2329	0.065
O3	104	2356		137	4	0.040
O2	22	18	186		82	0.071
O1	10	3	9	106		0.078

This shows that the fraction of detected transitions to the closed state versus to any state (red) is quite similar for each Ox state, with no consistent shift from O1 to O4.

3. ~Line 387. ‘...net negative charge of the conductive pore leads to higher K+ currents through it (although this may be challenged by recent MD simulations suggesting that conductance is inhibited if all four Kir2.2 D173 residues are ionized at lower membrane potentials [29].’ Based on other data in Kir2.1 it seems that the pronounced effects of protonation on conductance likely depend a lot on position, even in the inner cavity. I think that commonly used D172N or D172A mutations in the inner cavity of Kir2.1 do not have large effects on conductance, although they clearly affect the functional outcomes of the mutations (shown here) in the bundle crossing. This isn’t meant as a counterpoint to the data that are presented (which convincingly show variable protonation-dependent functional outcomes), but rather to consider that there may be important functional effects on properties other than conductance (such as blocker sensitivity).

We agree and have added some additional comment to note first that D172N or Q mutations do not have much effect on conductance in the native context, even though they dramatically affect inward rectification, illustrating the point above.

4. Line 258. ‘...we protonated one each of the four side chains of D173 and G178D in the second (Qcav -6) system. Could you clarify whether this was done in the same or different subunits? Or whether it matters? We have now clarified in which of the 4 subunits charges/neutralizations were made. Since the ions move in single file through the pore, it would seem highly unlikely that moving the G178D charges between subunits (relative to the subunits with the D173 charge) would have an effect, but we have not formally tested this.

5. Line 415. ‘... would be due to 0, 1, 2, or 3 (or 4) G178D residues being protonated’... I think this is a typo and likely means 177E in the context of the sentence.

The reviewer is correct. We have corrected the statement.

6. Perhaps consider mentioning work by Cymes and Grosman that illustrates protonation dependent side chains in the pore of nAChR channels, or perhaps other scenarios where acidic side chains undergo prominent pKa shifts.

We appreciate the extensive work of Cymes and Grosman in Cys-loop channels, illustrating protonation-dependent pore residues and the potential pKa shifts that can emerge, and we have now added additional reference to their work.

Reviewer #2

The authors studied Kir2.2 and Kir2.1 mutations at the inner helix bundle crossing which are known to open the inner gate and to have increased single channel conductances (SCC). These mutants (G178D and G178E in Kir2.2 and the homologous mutation in Kir2.1) displayed rather unusually stable subconductance levels, in comparison to those that are naturally occurring. The four subconductances that the authors observed for the mutants result from independent protonation of the acidic aspartates/glutamates in

each channel subunit. The protonation changes the K⁺ occupancy and the pore solvation at the helix bundle crossing and alters the G-gate. Noteworthy, while the SCC increases for WT Kir2.2 from 46 pS to 60 pS for the G178D mutant, the protonation of the mutant can result in a small open state subconductance with x0.45 of the maximal conductance, meaning 27 pS. The SCC which is smaller than that of WT channels, shows that the protonation does not reverse the gating phenotype of gain-of-function for the G178D mutation, but rather causes subconductance levels that are mechanistically different from the changes in gating observed for the G187D/E mutants. Thus, the conclusion in the Abstract and Introduction that gating and conductance are tightly coupled is somewhat correct, albeit limited only to the protonation state of these particular somewhat artificial mutant channels. The study investigates mutants to force the channel into an open state, also resulting in artificial subconductance levels. That the authors can titrate these subconductance levels by protonation and that it is understood how these subconductances can occur, is of biophysical interest, but does not foster our understanding of a naturally occurring gating mechanism, nor can there be any understanding derived for naturally occurring subconductance states of inward-rectifying Kir channels.

We appreciate the reviewer's careful reading of our paper. As noted in response to reviewer #1, we have now toned down the comments about conductance versus gating.

Regarding the artificial nature of the mutants, and the limitation of the interest to biophysics, we note that we are elucidating biophysical principles, not studying physiology. We are not seeking to provide understanding for naturally occurring Kir mutations, although the dbSNP database [<https://www.ncbi.nlm.nih.gov/snp>] does actually list naturally occurring human SNPs encoding G178D [rs1555562637], G178R and G178S [rs1555562636] in Kir2.2, as well as G177S [rs1555603949] and G177A [rs978533069] in Kir2.1.

-Abstract and discussion: the sentences with the "rectification controller" might need some re-writing sounds too much like lab slang.

We have revised this sentence and removed the term 'rectification controller' from the manuscript.

-The authors should discuss the limitation of their MDs with using 200 mM K⁺ and applying a voltage of 580 mV.

Since the simulation system is relatively large, and therefore the simulation timescales are forced to be relatively short, employing a higher voltage is state-of-the-art in MD simulations (now referenced), allowing detection of conductance throughout such simulations (e.g. <https://doi.org/10.1038/s41467-023-37531-8>; <https://doi.org/10.1085/jgp.201210820>). With respect to the ion concentration: this is within the range of experimentally utilized concentrations, and again is what is typically employed concentration in MD Simulations, although other groups have employed concentrations of up to 0.6M - 1M (e.g. <https://doi.org/10.1038/s41467-023-37531-8>; <https://doi.org/10.1085/jgp.201210820>).

-At page 7 Kir2.1 is stated to have a SCC of about 30 pS while at page 9 it is stated to have a SCC of about 25 pS.

We have corrected these statements for consistency.